# TITRATED: Learned Human Driving Behavior without Infractions via Amortized Inference

**Vasileios Lioutas**                                    *vasileios.lioutas@inverted.ai*
*Inverted AI*
*University of British Columbia*

**Adam Ścibior**                                         *adam.scibior@inverted.ai*
*Inverted AI*

**Frank Wood**                                           *frank.wood@inverted.ai*
*Inverted AI*
*University of British Columbia*
*Montréal Institute for Learning Algorithms (Mila)*

**Reviewed on OpenReview:** *https://openreview.net/forum?id=M8D5iZsnrO*

## Abstract

Models of human driving behavior have long been used for prediction in autonomous vehicles, but recently have also started being used to create non-playable characters for driving simulations. While such models are in many respects realistic, they tend to suffer from unacceptably high rates of driving infractions, such as collisions or off-road driving, particularly when deployed in map locations with road geometries dissimilar to the training dataset. In this paper we present a novel method for fine-tuning a foundation model of human driving behavior to novel locations where human demonstrations are not available which reduces the incidence of such infractions. The method relies on inference in the foundation model to generate infraction-free trajectories as well as additional penalties applied when fine-tuning the amortized inference behavioral model. We demonstrate this "titration" technique using the ITRA foundation behavior model trained on the INTERACTION dataset when transferring to CARLA map locations. We demonstrate a 76-86% reduction in infraction rate and provide evidence that further gains are possible with more computation or better inference algorithms.

## 1 Introduction

Predicting the behavior of human-operated vehicles is a crucial capability for self-driving cars. Traditional approaches are based on simple kinematic models (Cosgun et al., 2017; Ziegler et al., 2014; Lefèvre et al., 2014; Haarnoja et al., 2016) or hand-coded decision-making systems (Kesting et al., 2009), which may include parameters learned from data (Bhattacharyya et al., 2021). Recently data-driven approaches utilizing deep learning have become increasingly popular (Lee et al., 2017; Tang & Salakhutdinov, 2019; Djuric et al., 2020; Mo et al., 2020; Rhinehart et al., 2019; Gupta et al., 2018; Sadeghian et al., 2019; Zhao et al., 2020; Hong et al., 2019; Zhao et al., 2019).

Apart from their use in on-board prediction systems for planning in self-driving cars, models of human driving behaviors are also used to create realistic non-playable characters (NPCs) for simulators in which self-driving cars can be tested and trained (Suo et al., 2021; Bergamini et al., 2021). In both settings it is problematic if the models too-frequently generate behaviors that result in serious infractions, such as colliding with other agents or going off-road too often, as the resulting simulations would be sufficiently unrealistic to adversely impact downstream task performance. The problem of excess infractions can be to some extent mitigated

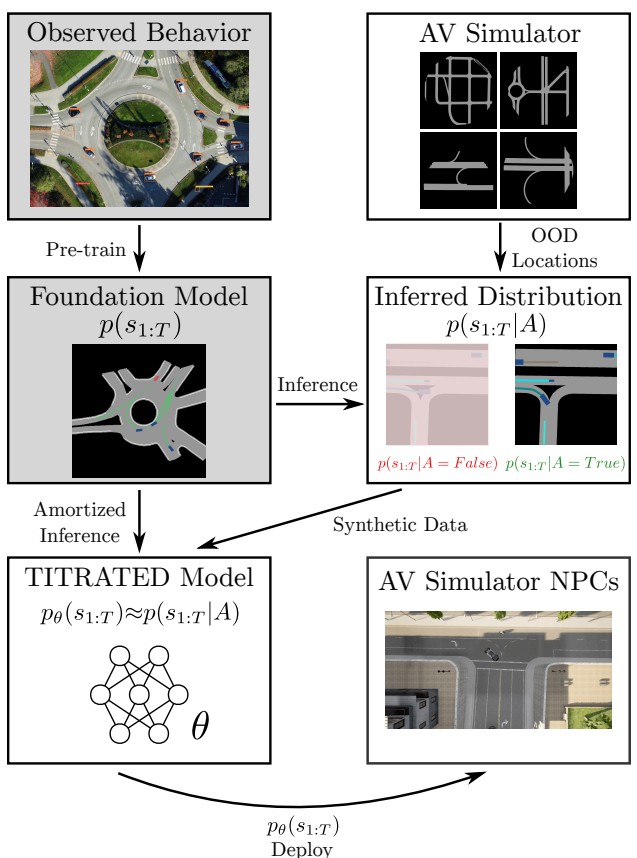

Figure 1: Overview of our model "titration" method. Starting from a pretrained foundation behavior model $p(s_{1:T})$ on observed human behavior (gray shaded boxes are taken as a given starting point), our aim is to transfer its predictive power to out of distribution (OOD) map geometries, particularly so as to use nearly-infraction-free behavior models to drive non-playable-characters (NPC)s in AV simulators. Our method is rooted in the use of inference to generate infraction-free trajectories in OOD locations from the conditional distribution $p(s_{1:T}|A)$ where $A$ is shorthand for the event "no infractions occur." By sampling behavioral rollouts from this distribution we generate synthetic data which is then, in turn, used to train a parametric "amortized" infraction-free behavior model $p_\theta(s_{1:T}) \approx p(s_{1:T}|A)$.

by introducing additional penalties for infractions at foundation behavior model training time, which are referred to as "common sense losses" in Suo et al. (2021).

Like all machine learning models, the ones predicting human driving behavior suffer from degraded performance under domain shift, which in this case, in particular, occurs when the models are deployed in locations not covered in the training dataset. This includes increased infraction rates, which is a major obstacle when creating simulations with NPCs learned from data collected on different roads. In this paper we present a general algorithm for fine-tuning behavior prediction models to novel settings where human demonstrations are not available.

Our key observation is that models predicting a distribution over possible future behaviors, rather than just a single behavior, often predict distributions that contain good predictions, but also others that perform infractions. Such predictions can be improved by further *conditioning* on not performing such infractions. While in certain settings such conditioning can be performed online during simulation (Suo et al., 2021), doing so is generally too computationally expensive.

We generate infraction-free trajectories in target locations from the conditional distribution defined by the predictive model via probabilistic inference offline. Subsequently, we use these trajectories to train a new

predictive model, typically by fine-tuning the original one, which targets the infraction-free conditional distribution. This effectively creates a synthetic dataset of behaviors that avoids infractions in the target locations, which can be used for learning behavior models through the usual means. This can be seen as a kind of highly structured data augmentation process. Alternatively, we can view this process as an instance of amortized inference, where the observations are known in advance, but the model is further conditioned, here on constraints. Figure 1 graphically illustrates our approach.

In this paper, we use ITRA (Ścibior et al., 2021) trained on the INTERACTION dataset (Zhan et al., 2019) as our foundation model (Bommasani et al., 2021) and target the creation of NPCs for CARLA (Dosovitskiy et al., 2017). We find that out-of-the-box ITRA generates trajectories that are in many ways realistic but suffer from unacceptably high rates of collisions and off-road driving on CARLA maps motivating the development of the method presented in this paper. We call the model output by applying our methodology TITRATED, which stands for "Training ITRA to Emulate Desiderata". The name is also loosely inspired by a form of "titration" we perform in Section 4.1 as we increase the number of rejection sampling attempts.

## 2 Related Work

Whether for producing onboard agent behavior predictions for autonomous vehicles (AV) or for generating realistic non-playable character (NPC) behaviors for AV simulators, behavior prediction models are at the heart of solving the AV control problem. The imitation of human driving behavior has been used to learn control policies for autonomous vehicles (Bojarski et al., 2016; Hawke et al., 2020). More recently, emphasis was put on probabilistic prediction models that behave more human-like and realistic (Suo et al., 2021; Bergamini et al., 2021; Ścibior et al., 2021). All machine learning data-driven behavior models suffer from generalizability issues, especially when the available training data is limited in the diversity of road structures and interesting agent interactions. To improve the performance of behavior models, Bansal et al. (2019) proposed using collision and off-road losses during training to reduce infraction rates. These losses are computed in the pixel space from the generated birdview representation masking the corresponding disallowed areas (i.e. the other agent and off-road parts). They complement the standard imitation loss that is used against human-recorded ground truth data, and thus, as training progresses, the model produces fewer and fewer infractions, leading the auxiliary losses to provide less information compared to the initial stages of training. On the other hand, Park et al. (2020) proposed to give as input to the probabilistic behavior model a probability heatmap that explicitly highlights the areas that are considered non-drivable. Zhu et al. (2021) suggested using an unlikelihood loss during training that will minimize the likelihood of undesired predictions possibly reducing the diversity of the predicted trajectories. Rhinehart et al. (2018) proposed R2P2 which is a single agent model that uses an autoregressive Gaussian at every step, that is conditionally dependent on all the previous steps. R2P2 emphasizes the use of a custom loss, which is a symmetrized cross-entropy loss between the model and data. This resembles a forward and reverse KL term. The latter is evaluated by rolling out under the model and evaluating an approximate data density, either a KDE estimate or a custom model scoring drivable surfaces trained separately. In contrast, in this work, we are interested in improving the performance of existing foundation probabilistic behavior models in OOD locations where acquiring human demonstration is difficult or even impossible. We amortize the inference process of generating infraction-free synthetic examples by finetuning with these examples and additionally applying collision and off-road losses by rolling out under the model during training.

## 3 Method

### 3.1 Foundation Model

Our starting point is ITRA (Ścibior et al., 2021) pre-trained on the INTERACTION dataset. This is the foundation model of driving behavior that we use in this work, but any other sufficiently general probabilistic behavior prediction model (e.g. (Zhao et al., 2019; Suo et al., 2021)) trained on any other dataset with sufficient coverage could equally well have been used. The method we develop is model-agnostic.

For convenience we share notation with ITRA. Let $s^i = \{s_1^i, \ldots, s_t^i, \ldots, s_T^i\}$ denote a sequence of states $s_t^i$ for agent $i$ where $i \in \{1, \ldots, N_t\}$ and $N_t$ is the number of agents in the scene at time $t$. Each agent state is a

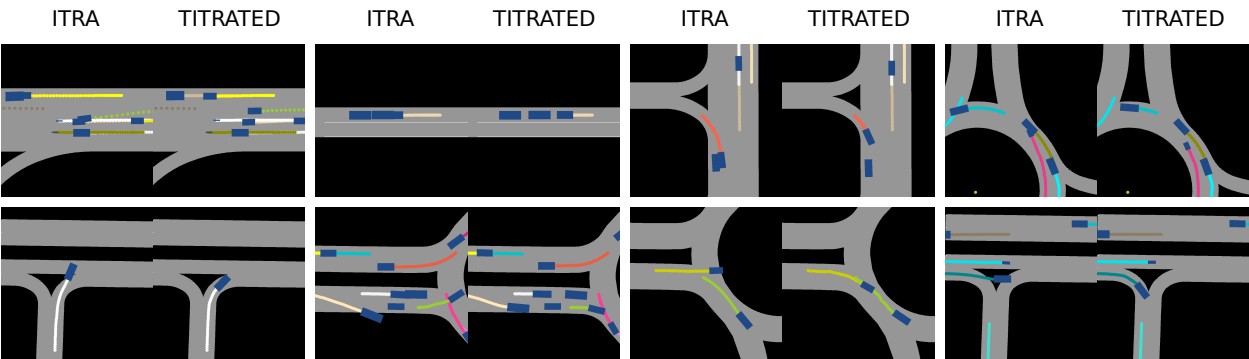

Figure 2: Examples from finetuning ITRA to the novel setting of CARLA. On its own ITRA produces unacceptable levels of infractions on CARLA maps, both in terms of collisions (top row) and off-road-driving (bottom row). Our method, TITRATED, substantially reduces both forms of infractions. Colored lines indicate the path moving vehicles took to get where they are.

tuple $s_t^i = (x_t^i, y_t^i, \psi_t^i, v_t^i) \in \mathbb{R}^4$ where $(x_t^i, y_t^i)$ is the geometric center of the agent, $\psi_t^i$ is the agent's orientation, and $v_t^i$ is its speed. Each agent has a fixed rectangular shape with length $l^i$ and width $w^i$, which is assumed known and implicitly a part of each agents' state.

In ITRA, the interactions of the agents with each other and the environment are mediated through a birdview image, which for each agent $i$ and time $t$ is obtained via differentiable rendering $\mathcal{I}_t^i = \texttt{render}(i, s_t^{1:N_t})$. These images are centered on each ego vehicle $i$ and rotated to match its orientation, but otherwise are similar to the images shown in Figure 2 albeit without the colored paths. Based on $\mathcal{I}_t^i$ and a hidden recurrent state, the agent $i$ at time $t$ selects action $a_t^i = (\alpha_t^i, \beta_t^i)$, consisting of acceleration and steering, which is translated into subsequent state $s_{t+1}^i$ using a bicycle kinematic model (Polack et al., 2017). This combination of birdview images and action space is selected to facilitate domain transfer of ITRA predictions to novel maps.

Each agent in ITRA is modeled as a conditional variational recurrent neural network (VRNN) (Chung et al., 2015), with latent variables $z_t^i$. The joint predictive distribution of ITRA factorizes as

$$p(s_{1:T}) = \prod_{t=1}^{T} \prod_{i=1}^{N_t} \int p(z_t^i) p(s_t^i | \mathcal{I}_t^i, z_t^i, h_{t-1}^i, s_{t-1}^i) \; dz_{1:T}^{1:N_t} \tag{1}$$

where

$$p(z_t^i) = \mathcal{N}(z_t^i; 0, \mathbf{I}) \tag{2}$$

$$p(s_{t+1}^i | \mathcal{I}_t^i, z_t^i, h_{t-1}^i, s_t^i) = \mathcal{N}(kin(s_t^i, a_t^i), \sigma\mathbf{I}) \tag{3}$$

$$h_{t+1}^i, a_t^i = \varphi(\mathcal{I}_t^i, z_t^i, h_{t-1}^i) \tag{4}$$

with $\varphi$ denoting the neural network and *kin* the kinematic bicycle model. The model is trained as usual, jointly with a separate inference network, by maximizing the evidence lower bound (ELBO). We refer the interested reader to (Ścibior et al., 2021) for more details.

## 3.2 Infractions

We start by formalizing the notion of infraction, notably collision and off-road invasion. We chose metrics that can indicate both the presence of such infractions and their severity. In Section 3.4 we use these metrics directly as losses in training.

For each agent $i$ and time $t$, the current state $s_t^i$ and the fixed dimensions $(l^i, w^i)$ define a bounding box $C_t^i \in \mathcal{R}^{4 \times 2}$, represented as four corners of the vehicle. The overlap between bounding boxes can be quantified

using the standard Intersection-over-Union (IoU) metric

$$\text{IoU}(B_i, B_j) = \frac{|B_i \cap B_j|}{|B_i \cup B_j|}. \tag{5}$$

Using that, we define our collision metric as the sum of individual IoUs across time

$$\mathcal{L}_{\text{C}}(C_{1:T}) = \sum_{t=1}^{T} \sum_{j \neq i}^{N_t} \text{IoU}(C_t^i, C_t^j). \tag{6}$$

As this metric is generally expensive to compute, particularly in a differentiable fashion, we use an approximate version of it described in Zhou et al. (2019). Contrary to other methods (Suo et al., 2021; Rempe et al., 2022) that employ a differentiable collision loss that approximates each vehicle as 5 circles and maximizes the distance between the closest centroids, our use of the technique of Zhou et al. (2019) is novel in that it is a highly accurate differentiable collision loss that directly minimizes the overlapping area between two collided vehicles.

For off-road infractions, we assume that we have access to a triangle mesh $V$ defining the driveable area (road surfaces, etc.). Given a function $\Phi$ that computes the distance from a point to a triangle, we define a novel off-road infraction metric as the sum of distances of the corners of all vehicles from the road mesh across time

$$\mathcal{L}_{\text{OR}}(C_{1:T}, V) = \sum_{t=1}^{T} \sum_{i=1}^{N_t} \sum_{c_j \in C_t^i} \min_{v \in V} \Phi(c_j, v). \tag{7}$$

This metric can be computed efficiently in a differentiable manner (Eberly, 1999) and well-established implementations exist.[1] When used in training, this metric is referred to as the point-to-surface loss (Smith et al., 2019). Note that it is zero only when all four corners of all vehicles are contained within the driveable area.

### 3.3 Inference

Let $A$ be the event that no vehicle performed an infraction within the specified time window

$$A := (\mathcal{L}_{\text{C}} = 0) \wedge (\mathcal{L}_{\text{OR}} = 0) \tag{8}$$

where $\wedge$ means logical and. The probability of $A$ under the real-world distribution of human driving behaviors is remarkably close to 1, but in predictive models such as ITRA it can be much lower, indicating an unacceptable, unrealistic number of infractions. Our key insight is that we can obtain a better model by further conditioning on not performing infractions, denoted here as the conditional density $p(s_{1:T}|A, V)$. Note that in order to keep the notation relatively uncluttered we have throughout deliberately elided notation that indicates the conditional dependence on the underlying map layout $V$. It should be kept in mind throughout that all of our objectives are averaged over $V$.

Unfortunately, sampling from the conditional distribution $p(s_{1:T}|A)$ can be computationally expensive. This is particularly problematic in the simulation context, where we would normally unroll the predictive model one step at a time to derive NPC behaviors, but conditioning on not performing infractions in the future requires also sampling subsequent states. For this reason, in Section 3.4 we learn a model $p_\theta(s_{1:T}) \approx p(s_{1:T}|A)$, which can be sampled from sequentially in fixed time. First, we show how to use inference to generate examples of infraction-free trajectories in an off-line setting.

Given a collection $D$ of initial conditions on target maps, we can construct a synthetic dataset $\tilde{D}$ of infraction-free trajectories by repeatedly sampling joint trajectories from the predictive model and rejecting them until no infractions are found. This procedure is simple but can be computationally expensive, since rejection sampling can take arbitrarily long to produce acceptable samples. In order to limit computational cost, we introduce a *max_trials* parameter, which indicates the maximum number of sampling attempts performed,

---

[1] https://github.com/NVIDIAGameWorks/kaolin

---

**Algorithm 1** Infraction-Free Dataset Generation

---

**Input:** Initial conditions dataset $D$
  Driving behaviour model $p(s_{1:T})$
  Maximum number of trials $max\_trials$
**Output:** Infraction-Free dataset $\tilde{D}$

1: $\tilde{D} \leftarrow \emptyset$
2: **for** $(s_0, V) \in D$ **do**
3:      $found \leftarrow$ **false**
4:      **for** $n \leftarrow 1$ to $max\_trials$ **do**
5:          Sample rollout $s_{1:T}$ from $p(s_{1:T}|s_0)$
6:          Convert states $s_{1:T}$ to bounding boxes $C_{1:T}$
7:          **if** $\mathcal{L}_C(C_{1:T}) = 0$ and $\mathcal{L}_{OR}(C_{1:T}, V) = 0$ **then**
8:             $found \leftarrow$ **true**
9:             **break**
10:      **if** $found$ **then**
11:          $\tilde{D} \leftarrow \tilde{D} \cup \{(s_{1:T}, V)\}$
12: **return** $\tilde{D}$

---

after which the $(s_0, V)$ tuple is excluded from the dataset if an acceptable sample was not found. The details of this dataset generation process are given in Algorithm 1. Note that this process can be done "just-in-time", interleaved with training, which is what we actually do in our experiments. This has the advantage of not allowing the fine-tuned model to overfit to specific examples and requires, instead, it to fit the full conditional distribution.

### 3.4 Amortized Inference

Using the synthetic data consisting of infraction-free trajectories from Section 3.3, we can train an amortized, "titrated", model with ELBO, following the standard training process of ITRA outlined in Section 3.1. This process minimizes the negative ELBO, defined as

$$
\mathcal{L}_{\text{ELBO}} = \mathop{\mathbb{E}}_{s_{1:T} \sim p(s_{1:T}|A)} \left[ \sum_{t=1}^{T-1} \sum_{i=1}^{N_t} \left( \mathop{\mathbb{E}}_{q_\phi(z_t^i|a_t^i, \mathcal{I}_t^i, h_t^i)} \left[ \log p_\theta(s_{t+1}^i | \mathcal{I}_t^i, z_t^i, h_t^i) \right] - D_{\text{KL}} \left[ q_\phi(z_t^i|a_t^i, \mathcal{I}_t^i, h_t^i) || p_\theta(z_t^i) \right] \right) \right]
$$
$$
\leq \mathop{\mathbb{E}}_{s_{1:T} \sim p(s_{1:T}|A)} \left[ \log p_\theta(s_{1:T}) \right] \tag{9}
$$

where $q_\phi$ is a separate inference network trained jointly with the model $p_\theta$.

As discussed in Section 1, learned models of human driving behavior, such as ITRA, are prone to generate excessive infractions. We ameliorate this problem by introducing explicit infraction penalties in the form of the collision metrics from Section 3.2, obtained when sampling from the amortized model $p_\theta$, as additional loss terms

$$
\mathcal{L} = -\mathcal{L}_{\text{ELBO}} + \lambda_C \mathcal{L}_C + \lambda_{OR} \mathcal{L}_{OR}. \tag{10}
$$

This is the final training objective for TITRATED, although some care needs to be taken when minimizing it, since $\mathcal{L}_{\text{ELBO}}$ involves sampling from the inference network, while $\mathcal{L}_C$ and $\mathcal{L}_{OR}$ involve sampling from the prior. We address this by performing two separate rollouts for each training example, as shown in detail in Algorithm 2.

## 4 Experimental Results

In our experiments, we, at a high level, learn human driving behaviors from the INTERACTION dataset (Zhan et al., 2019) and use them to create NPCs in CARLA (Dosovitskiy et al., 2017). More specifically we take a pre-trained ITRA model obtained exactly as described in Ścibior et al. (2021) and fine-tune it

---

**Algorithm 2** Amortized Inference with Infraction Penalties Training Process

---

**Input:** Infraction-Free dataset $\tilde{D}$ (Alg 1)
        Foundation model $p$
**Output:** Amortized model $p_\theta$

1: Initialize $p_\theta$ as $p$
2: **for** $(s_{1:T}, V) \in \tilde{D}$ **do**
3:     $\hat{s}_{1:T-1} \sim p_\theta(s_{1:T})$ using proposal distribution $q_\phi$
4:     $\mathcal{L}_{\text{ELBO}} \leftarrow$ compute using $s_{1:T}$ and $\hat{s}_{1:T-1}$ (Eq 9)
5:
6:     $\tilde{s}_{1:T-1} \sim p_\theta(s_{1:T})$ using prior distribution
7:     Convert states $\tilde{s}_{1:T-1}$ to bounding boxes $\tilde{C}_{1:T-1}$
8:     $\mathcal{L}_{\text{C}} \leftarrow$ compute using $\tilde{C}_{1:T-1}$ (Eq 6)
9:     $\mathcal{L}_{\text{OR}} \leftarrow$ compute using $\tilde{C}_{1:T-1}$ and $V$ (Eq 7)
10:
11:     $\mathcal{L} \leftarrow -\mathcal{L}_{\text{ELBO}} + \lambda_{\text{C}}\mathcal{L}_{\text{C}} + \lambda_{\text{OR}}\mathcal{L}_{\text{OR}}$ (Eq 10)
12:     $\theta \leftarrow \theta - \eta\frac{\partial\mathcal{L}}{\partial\theta}$
13: **return** $p_\theta$

---

Table 1: Summary of the dataset collected using CARLA autopilot. These are the initial conditions used to adapt TITRATED to CARLA.

| Scene | Train / Val. Size | Avg. Num. Agents Per Trajectory |
|---|---|---|
| Town01_Straight | 7338 / 492 | 8.6 |
| Town01_3way | 18632 / 930 | 19.8 |
| Town02_Straight | 36215 / 1916 | 35.5 |
| Town02_3way | 38206 / 2030 | 38.8 |
| Town03_Roundabout | 21830 / 1151 | 24.1 |
| Town03_5way | 22160 / 1076 | 24.1 |
| Town03_4way | 12822 / 753 | 13.8 |
| Town03_3way_Unprotected | 14865 / 885 | 15.0 |
| Town03_3way_Protected | 13045 / 688 | 14.5 |
| Town03_GasStation | 21127 / 1028 | 21.2 |
| Town04_Merging | 5268 / 292 | 7.3 |
| Town04_3way_Large | 3931 / 193 | 5.3 |
| Town04_3way_Small | 5076 / 394 | 6.2 |
| Town04_4way_Stop | 7872 / 582 | 8.5 |
| Town04_Parking | 12794 / 701 | 13.8 |
| Town06_Merge_Single | 13154 / 664 | 16.5 |
| Town06_4way_large | 9110 / 362 | 9.7 |
| Town06_Merge_Double | 5028 / 240 | 7.5 |
| Town07_3way | 35601 / 1871 | 37.0 |
| Town07_4way | 30412 / 1586 | 31.8 |
| Town10HD_4way | 57826 / 3005 | 55.9 |
| Town10HD_3way_Protected | 30456 / 1485 | 30.0 |
| Town10HD_3way_Stop | 50747 / 2566 | 50.0 |

with the method described in Section 3 to obtain TITRATED models for a collection of selected locations in CARLA. Our goal is to maintain maximum similarity to ITRA predictions while reducing the incidence of driving infractions, consisting of collisions and off-road invasions, in these novel simulator contexts.

To create a collection of initial conditions required by Algorithm 1 in CARLA, we created a custom synthetic dataset using the built-in autopilot. Specifically, in each town we recorded 200 minutes of driving at 10 frames per second, featuring 100 randomly selected and placed vehicles. We then selected 23 locations of interest

Table 2: Rates of driving infractions in various CARLA scenes, with (TITRATED) and without (ITRA) the fine-tuning proposed in this paper. TITRATED (All maps) refers to a single model trained on all 23 distinct locations.

| Scene | Collision Rate $\times 10^{-4}$ | | | | Off-road Rate $\times 10^{-4}$ | | | |
|---|---|---|---|---|---|---|---|---|
| | ITRA | TITRATED | TITRATED (All maps) | Reduction | ITRA | TITRATED | TITRATED (All maps) | Reduction |
| Town01_Straight | 26.0 | **2.0** | 5.6 | 78.46% | 42.7 | **0** | **0** | 100% |
| Town01_3way | 19.4 | 4.8 | **4.3** | 77.84% | 11.3 | **0** | **0** | 100% |
| Town02_Straight | 11.2 | **2.8** | 3.4 | 69.64% | 51.5 | **1.6** | 4.9 | 90.49% |
| Town02_3way | 10.0 | 3.3 | **3.1** | 69.00% | 33.9 | **0** | 0.5 | 98.53% |
| Town03_Roundabout | 5.0 | 1.7 | **1.2** | 76.00% | 134.0 | 47.1 | **7.4** | 94.48% |
| Town03_5way | 4.6 | 1.6 | **0.9** | 80.43% | 95.8 | 37.5 | **3.7** | 96.14% |
| Town03_4way | 6.0 | 2.5 | **1.3** | 78.33% | 67.2 | 34.2 | **19.6** | 70.83% |
| Town03_3way_Unprotected | 6.3 | 2.7 | **2.6** | 58.73% | 74.8 | 42.3 | **25.1** | 66.44% |
| Town03_3way_Protected | 10.4 | 4.9 | **3.9** | 62.50% | 84.9 | 41.5 | **27.4** | 67.73% |
| Town03_GasStation | 6.6 | 0.6 | **0.3** | 95.45% | 32.8 | **13.2** | 19.1 | 41.77% |
| Town04_Merging | 0.2 | 0.1 | **0** | 100% | 47.8 | **0** | **0** | 100% |
| Town04_3way_Large | 8.0 | 2.3 | **1.4** | 82.5% | 11.2 | **0** | **0** | 100% |
| Town04_3way_Small | 16.6 | 6.4 | **3.4** | 79.52% | 29.6 | 0.8 | **0** | 100% |
| Town04_4way_Stop | 10.9 | 4.7 | **2.0** | 81.65% | 32.4 | 11.6 | **0.09** | 99.72% |
| Town04_Parking | 6.5 | 4.7 | **1.1** | 83.08% | 44.7 | 5.3 | **0.1** | 99.78% |
| Town06_Merge_Single | 1.7 | **1.1** | 1.3 | 23.53% | 109.6 | 27.6 | **20.0** | 81.75% |
| Town06_4way_large | 1.7 | 0.3 | **0** | 100% | 7.3 | **1.5** | 3.3 | 54.79% |
| Town06_Merge_Double | 1.8 | 0.5 | **0** | 100% | 2.3 | **0** | **0** | 100% |
| Town07_3way | 6.9 | 2.2 | **2.0** | 71.01% | 45.3 | 3.1 | **1.0** | 97.79% |
| Town07_4way | 7.8 | **2.3** | 2.6 | 66.67% | 60.6 | 3.1 | **0.1** | 99.83% |
| Town10HD_4way | 3.6 | 1.7 | **1.0** | 72.22% | 51.1 | 21.1 | **10.9** | 78.67% |
| Town10HD_3way_Protected | 4.5 | 1.6 | **0.9** | 80.00% | 76.2 | 26.3 | **21.0** | 72.44% |
| Town10HD_3way_Stop | 3.6 | 1.9 | **1.3** | 63.89% | 62.5 | 17.2 | **12.8** | 79.52% |
| Average | 7.8 | 2.4 | **1.8** | 76.11% | 52.6 | 14.5 | **7.6** | 86.55% |

across 7 towns and cropped the recordings to a 100 meter radius from each of them, mimicking the structure of the INTERACTION dataset. We then sliced those datasets into 4 second segments, consisting of 1 second of history and 3 seconds of predictions, taking the 1 second history from those segments as initial conditions. The dataset is summarized in Table 1. We emphasize that the autopilot-generated trajectories are only used to specify initial conditions, the trajectories recorded in the subsequent 3 seconds are never used in training.

In the evaluation we are only interested in the frequency of infractions and not their severity, so we define infraction rates using indicator functions on the metrics defined in Section 3.2. Specifically, we define the collision rate as

$$\text{Collision Rate} = \frac{1}{N}\frac{1}{K}\frac{1}{T}\sum_{n=1}^{N}\sum_{k=1}^{K}\sum_{t=1}^{T}\mathbb{1}_{\mathcal{L}_{\text{C}}(s_{n,t}^{(k)})>0} \tag{11}$$

where $N$ is the number of validation examples and $K$ is the number of samples generated for each example $n \in N$. Contrary to other methods (Suo et al., 2021), we apply zero collision tolerance since our method is able to detect collisions precisely instead of approximately. We define the off-road rate as

$$\text{Off-road Rate} = \frac{1}{N}\frac{1}{K}\frac{1}{T}\sum_{n=1}^{N}\sum_{k=1}^{K}\sum_{t=1}^{T}\mathbb{1}_{\mathcal{L}_{\text{OR}}(s_{n,t}^{(k),i})>0} \tag{12}$$

where $i$ refers to the index of the ego-vehicle.

In each of the CARLA locations, we produce a separate version of TITRATED by fine-tuning ITRA from the checkpoint trained on the INTERACTION dataset. In addition, we train a single TITRATED model for all CARLA locations. We use ITRA (Ścibior et al., 2021) as our baseline model pre-trained on the whole INTERACTION dataset (Zhan et al., 2019). We follow the same hyper-parameter setup as described in Ścibior et al. (2021) and fine-tune a mirror copy of the ITRA model for each unseen location from the

Table 3: The effect of more effective inference on TITRATED infraction rates.

| Scene | Max Trials | | | | | | | | | | | | | | |
|---|---|---|---|---|---|---|---|---|---|---|---|---|---|---|---|
| | 1 | 10 | 20 | 50 | 100 | 1 | 10 | 20 | 50 | 100 | 1 | 10 | 20 | 50 | 100 |
| | Collision Rate $\times 10^{-4}$ | | | | | Off-road Rate $\times 10^{-4}$ | | | | | Rejected Examples | | | | |
| Straight Road | 11.21 | 2.09 | 2.07 | 2.02 | 1.58 | 2.25 | 0 | 0 | 0 | 0 | 7.5% | 2.6% | 2.1% | 1.7% | 1.5% |
| Roundabout | 3.45 | 1.72 | 1.72 | 1.44 | 1.43 | 63.32 | 47.15 | 37.88 | 35.18 | 34.60 | 10.1% | 5.4% | 4.8% | 4.1% | 3.7% |
| 4-way Intersection | 1.05 | 0.34 | 0.30 | 0 | 0 | 2.60 | 1.53 | 1.38 | 0.92 | 0.46 | 2.8% | 1.3% | 1.2% | 1.0% | 0.9% |

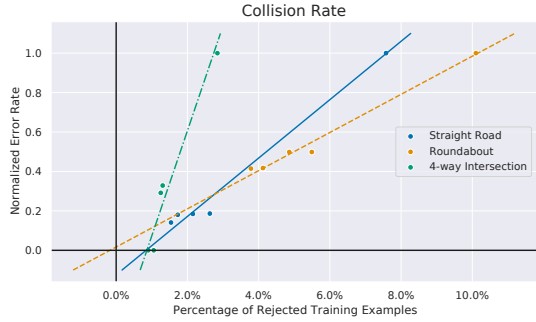
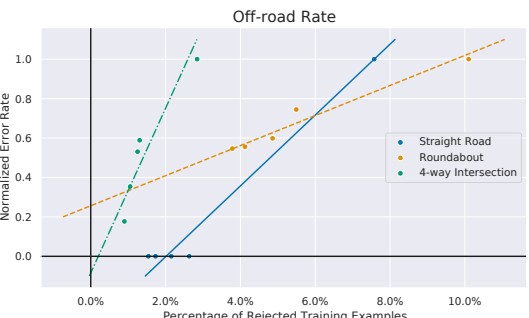

Figure 3: Scatter plots of infraction rates achieved by TITRATED versus the number of items rejected from the synthetic dataset. Different points correspond to $max\_trials$ values $\in \{1, 10, 20, 50, 100\}$ used in rejection sampling, increasing to the left. Straight lines are fit with least squares to extrapolate performance to a setting with perfect inference algorithms where no items are rejected.

synthetic dataset using the proposed methodology. Specifically, we set the model hidden size $d_h$ to 64, the latent representation size $d_z$ to 2 and the number of recurrent layers to 2. We use differentiable rendering as described in Ścibior et al. (2021) and a birdview image of size 256×256. We use the ADAM optimizer (Kingma & Ba, 2015) with default values and $1 \times 10^{-3}$ learning rate. The batch size is set to 8 trajectories using class-mates forcing (Ścibior et al., 2021; Tang & Salakhutdinov, 2019) and trained on a single NVIDIA RTX 2080 Ti GPU. We set the hyperparameters $\lambda_C$ and $\lambda_{OR}$ to $1 \times 10^3$ and $1 \times 10^2$ respectively and use $max\_trials$ of 10 and just-in-time sampling of infraction-free training examples.

Table 2 shows that TITRATED is able to significantly reduce the collision and off-road rates of ITRA. Specifically, we first generate an ITRA rollout for each initial condition in the corresponding CARLA dataset (see Table 1) and compute infraction rates. We do the same for TITRATED rollouts. We see that, averaged across all CARLA locations, individually trained TITRATED models are able to reduce collisions by 69% and off-road invasions by 72%. In some cases, TITRATED even achieves a full 100% reduction, entirely eliminating off-road driving in all validation examples. Furthermore, our single TITRATED model trained on all CARLA locations is producing a 76% reduction in collisions and 86% in off-road infractions. Figure 3 and the related discussion in the following section suggest that TITRATED is likely to reduce infractions even further with increased quality of inference, albeit at potentially greater computational expense to construct infraction-free examples for all initial conditions.

## 4.1 Better Inference Leads to Lower Infraction Rates

The key component of TITRATED is the Bayesian inference algorithm for computing and sampling from the conditional distribution $p(s_{1:T}|A)$. In the main experiment, we used rejection sampling with $max\_trials = 10$, resulting in a reasonable trade-off between computational cost and infraction metric improvements. However, this choice excludes many examples for which no accepted sample could be found in ten tries. Crucially, these are "difficult" examples in the sense that ITRA is likely to produce high infraction rates on them. This raises the following question: would finding accepted samples for all of these difficult examples drive TITRATED model infraction rates to zero?

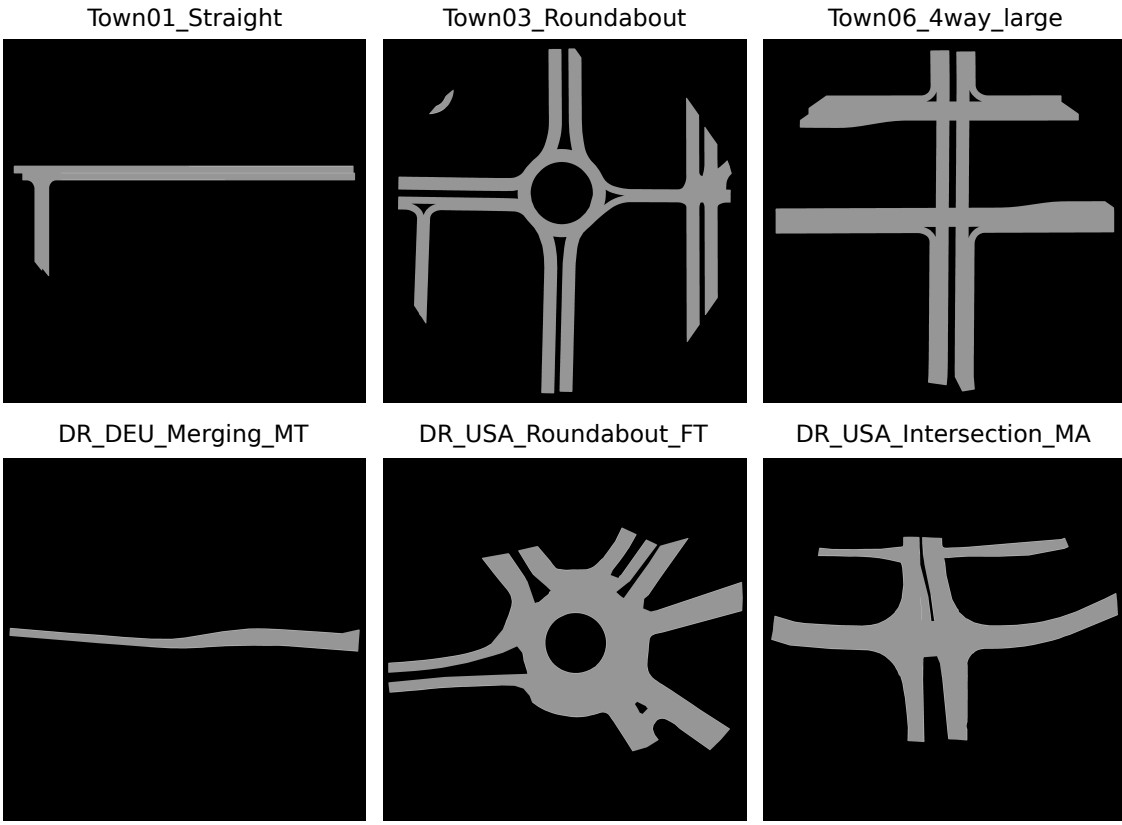

Figure 4: Top row: three representative scenes from our CARLA dataset; a straight road, a roundabout, and a 4-way intersection. Bottom row: the structurally similar maps from the INTERACTION dataset that we used for the acceleration distribution comparison in Figure 5.

To answer this question, we investigated how using increasingly more powerful inference algorithms could impact the performance of TITRATED, trying to extrapolate to the setting of a perfect inference oracle that is able to produce samples with no infractions in all cases. For simplicity, we conducted this study using rejection sampling with increasing $max\_trials$ to stand in for a sequence of increasingly powerful inference algorithms, although in practice one could use a different sequence of sampling techniques of growing sophistication or computational expense instead.

We selected three representative locations from the CARLA dataset, covering a straight road, a roundabout, and a 4-way intersection, as depicted in Figure 4. For each of those locations, we ran the full TITRATED training procedure with varying values of $max\_trials \in \{1, 10, 20, 50, 100\}$. As expected, both the number of rejected examples and the infraction rate decrease monotonically with increasing $max\_trials$, as shown in Table 3. This procedure of gradually increasing the number of rejection sampling attempts loosely resembles titration, further motivating the naming of our method.

To analyze the scaling behavior as the inference algorithm improves, we plotted the infraction rate of TITRATED against the fraction of rejected examples in Figure 3. Somewhat surprisingly, this scaling tends to be close to linear, allowing us to extrapolate to a setting with a perfect inference algorithm that finds acceptable samples for all training examples. In most, but not all cases, the extrapolated line achieves zero infraction rate before achieving full dataset coverage, suggesting that the use of better inference algorithms could remove infractions entirely.

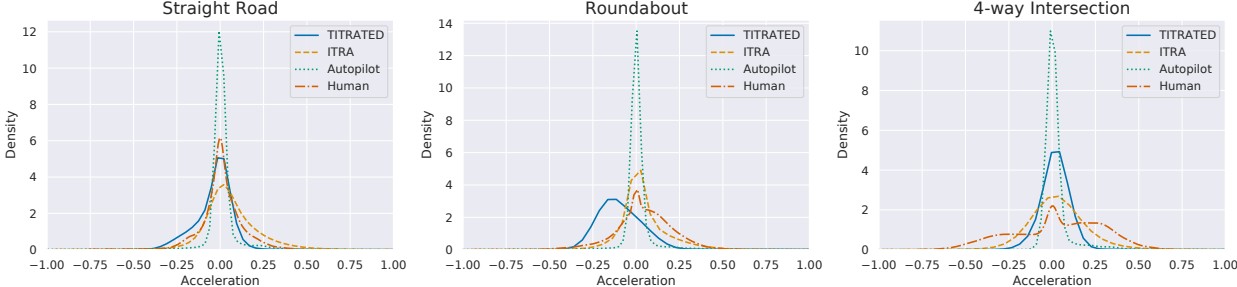

Figure 5: Kernel density estimates of acceleration distributions for TITRATED, ITRA, CARLA autopilot, and real human data. Acceleration values are normalized using highest values present in human data.

## 4.2 Assessing Human-likeness of Driving Behavior

The primary motivation for this work was to create NPCs for CARLA exhibiting human-like driving behavior. While this is largely subjective and no definitive metrics are agreed-upon by the community, in this experiment we seek to demonstrate that in some sense, TITRATED indeed generates more human-like driving behavior than the CARLA autopilot.

When actual human trajectories for specific maps are available, prediction metrics such as average and final displacement error can be used to judge the realism of behavior models (Bergamini et al., 2021), but unfortunately those are not available in CARLA. For comparing different learning-based models, infraction rates can be used, but hand-coded models such as CARLA autopilot can easily achieve zero infraction rates while still not being very realistic.

The only alternative approach we are aware of is to compare the distributions of various quantities associated with driving (Chao et al., 2020). In Figure 5 we present the probability densities of vehicle acceleration. We chose three representative scenes from the CARLA dataset, the same ones as used in Section 4.1, and selected similar scenes from the INTERACTION dataset for comparison with humans in a similar scenario. The scenes are depicted in Figure 4.

In all three cases, CARLA autopilot has a highly peaked acceleration distribution around zero, while the distribution of human acceleration values is more spread, corresponding to a higher diversity of controls applied by humans. We see that both ITRA and TITRATED are more realistic than the CARLA autopilot, in the sense of the distribution of accelerations being more spread. As might be expected, the additional conditioning in TITRATED sometimes leads to less realistic acceleration distributions, reducing the variance on 4-way Intersection and introducing excessive braking on Roundabout. On the other hand, on Straight Road the TITRATED distribution is slightly less peaked than ITRA, indicating a more realistic acceleration distribution after conditioning on not performing infractions.

## 5 Discussion

A limitation of TITRATED is that it can only be used to address infractions that we never want to see in predictions. However, humans commit those infractions in the real world at a non-zero rate, and a realistic model should be able to replicate this rate of infractions. This is even more important for less catastrophic infractions, such as running a red light or failing to yield. Our proposed methodology for amortized conditional inference is agnostic to the type and number of conditions. In principle, one can use more sophisticated conditioning to achieve such goals. A promising avenue for future work is accommodating soft constraints that we expect humans to satisfy usually, but not always, like obeying traffic rules or executing specific maneuvers and commands. Taking advantage of more powerful interference algorithms is essential for achieving this goal, thus we are currently investigating alternatives in this context.

## 6    Conclusion

TITRATED provides a way to transfer learned models of human driving behaviors to new settings while avoiding unrealistic infractions typically associated with such domain transfer. The crucial ingredient of the process is generating samples from the model conditioned on not performing such infractions, which is challenging, but better inference algorithms used for this purpose may result in large performance improvements. TITRATED is already used to provide NPCs for CARLA based on human driving trajectories recorded in the real world.

### Acknowledgments

We acknowledge the support of the Natural Sciences and Engineering Research Council of Canada (NSERC), the Canada CIFAR AI Chairs Program, and the Intel Parallel Computing Centers program. Additional support was provided by UBC's Composites Research Network (CRN), and Data Science Institute (DSI). This research was enabled in part by technical support and computational resources provided by WestGrid (www.westgrid.ca), Compute Canada (www.computecanada.ca), and Advanced Research Computing at the University of British Columbia (arc.ubc.ca).

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
