# OpenReview forum: "TITRATED: Learned Human Driving Behavior without Infractions via Amortized Inference"
_TMLR — Accepted by TMLR_

### Review · Reviewer_TsWW · 2022-07-01

**Summary Of Contributions:**

This paper proposes a method for vehicle motion forecasting that incorporates feedback from infraction metrics to build a finetuned model. The method is motivated from the perspective of learning an amortized inference model. An experimental evaluation in CARLA shows several things:

- 1) relative to the non-finetuned model, the infraction metrics are much lower (on average, 76% reduction in collision rate, and 87% reduction in off-road rate).
- 2) increasing the amount of samples considered during rejection sampling results in better performance, which the paper then claims could be extrapolated to hold for other methods that lead to better approximate inference.
- 3) The acceleration profiles of various methods in a few difference scenes show that the proposed method and the method upon which it is based do better than the CARLA autopilot in matching the distribution of human accelerations.

**Requested Changes:**

Please address the three significant weaknesses, as I see all as critical to the acceptability of the paper.

# Updated requested changes.
My requested changes have been implemented, so none remain.

**Strengths And Weaknesses:**

# Strengths summary
+ The performance evaluation w.r.t. to the prior method is sufficient for us to confidently conclude a significant and substantial performance improvement.
+ The broader goal of building 'NPC' agents for CARLA / other simulators is worthwhile and of interest to the community.

# Weaknesses summary
There are several important weaknesses.
- This paper is missing discussion of and comparison to related work (there's no separate related work section, which isn't ideal). There are closely related existing methods core to the proposed method: there exists methods for incorporating sample infraction information into the training objective for vehicle motion prediction, including [A, B, C, D]. At the very least, verbal comparison to these related works is necessary, in order for the paper to be clear about its contributions.
- This paper is missing a baseline in which there is no separation between the models p(s|A) and p(s), but, unlike ITRA, also uses infraction metrics, but unlike the proposed method, just uses them to train p(s). Perhaps this isn't possible given the nature of the pretraining dataset, but I would like the authors to consider and comment.
- This paper is missing a clear learning algorithm that uses the data generated by Alg 1 in the objective function (Eq 10). Therefore, the proposed method itself is unclear.

# Updated weaknesses summary
My weakness concerns have all been resolved following the revisions and the discussion in the author response thread attached to this review.

## Related work
- [A] "ChauffeurNet: Learning to Drive by Imitating the Best and Synthesizing the Worst" https://arxiv.org/abs/1812.03079v1 arXiv 2018
- [B] "R2p2: A reparameterized pushforward policy for diverse, precise generative path forecasting" ECCV 2018 https://link.springer.com/chapter/10.1007/978-3-030-01261-8_47
- [C] "Diverse and admissible trajectory forecasting through multimodal context understanding" ECCV 2020 https://link.springer.com/chapter/10.1007/978-3-030-58621-8_17
- [D] "Motion Forecasting with Unlikelihood Training in Continuous Space" CoRL 2021 https://openreview.net/forum?id=4u25M570Iji

---

> ### Author Response · Authors · 2022-07-18
> **Response to Reviewer TsWW**
>
> We would like to thank the reviewer for their feedback. We added a related work section in the revised manuscript.
>
> - **Regarding “This paper is missing a baseline in which there is no separation between the models p(s|A) and p(s), but, unlike ITRA, also uses infraction metrics, but unlike the proposed method, just uses them to train p(s).”**
>
> For this experiment, we trained an ITRA model on the INTERACTION dataset by additionally applying the proposed infraction losses and measured the infraction rates by deploying it on the OOD CARLA locations.
>
> |  | **Collision Rate** ${\times}10^{-4}$ | **Collision Rate** ${\times}10^{-4}$ | **Collision Rate** ${\times}10^{-4}$ | **Off-road Rate** ${\times}10^{-4}$ | **Off-road Rate** ${\times}10^{-4}$ | **Off-road Rate** ${\times}10^{-4}$ |
> |---|---|---|---|---|---|---|
> | **Scene name** | **ITRA** | **ITRA (With infraction losses)** | **TITRATED** | **ITRA** | **ITRA (With infraction losses)** | **TITRATED** |
> | Town01_Straight | 26 | 14.3 | 5.6 | 42.7 | 0.9 | 0 |
> | Town01_3way | 19.4 | 12 | 4.3 | 11.3 | 12.5 | 0 |
> | Town02_Straight | 11.2 | 8.2 | 3.4 | 51.5 | 14.1 | 4.9 |
> | Town02_3way | 10 | 9.1 | 3.1 | 33.9 | 15.2 | 0.5 |
> | Town03_Roundabout | 5 | 4.2 | 1.2 | 134 | 84.4 | 7.4 |
> | Town03_5way | 4.6 | 3.5 | 0.9 | 95.8 | 62.2 | 3.7 |
> | Town03_4way | 6 | 3.9 | 1.3 | 67.2 | 45.1 | 19.6 |
> | Town03_3way_Unprotected | 6.3 | 4.9 | 2.6 | 74.8 | 79.8 | 25.1 |
> | Town03_3way_Protected | 10.4 | 7.9 | 3.9 | 84.9 | 72.3 | 27.4 |
> | Town03_GasStation | 6.6 | 3.6 | 0.3 | 32.8 | 30.2 | 19.1 |
> | Town04_Merging | 0.2 | 0.1 | 0 | 47.8 | 32.7 | 0 |
> | Town04_3way_Large | 8 | 5.8 | 1.4 | 11.2 | 1.4 | 0 |
> | Town04_3way_Small | 16.6 | 11.3 | 3.4 | 29.6 | 18.1 | 0 |
> | Town04_4way_Stop | 10.9 | 7.7 | 2 | 32.4 | 28.5 | 0.09 |
> | Town04_Parking | 6.5 | 5.8 | 1.1 | 44.7 | 66.4 | 0.1 |
> | Town06_Merge_Single | 1.7 | 0.07 | 1.3 | 109.6 | 80.5 | 20 |
> | Town06_4way_large | 1.7 | 2.1 | 0 | 7.3 | 15.1 | 3.3 |
> | Town06_Merge_Double | 1.8 | 2 | 0 | 2.3 | 2.7 | 0 |
> | Town07_3way | 6.9 | 5.4 | 2 | 45.3 | 63.6 | 1 |
> | Town07_4way | 7.8 | 5.7 | 2.6 | 60.6 | 40.7 | 0.1 |
> | Town10HD_4way | 3.6 | 2.8 | 1 | 51.1 | 47.5 | 10.9 |
> | Town10HD_3way_Protected | 4.5 | 3.7 | 0.9 | 76.2 | 53.4 | 21 |
> | Town10HD_3way_Stop | 3.6 | 3.6 | 1.3 | 62.5 | 47.9 | 12.8 |
> | Average | 7.8 | 5.5 | 1.8 | 52.6 | 39.7 | 7.6 |
>
> Training ITRA with the infraction losses helps reduce the infraction rates when deployed on OOD locations. Still, the generalizability issue remains highlighting the need to apply the TITRATED method for better performance in OOD locations.

---

> > ### Comment · Reviewer_TsWW · 2022-07-20
> > **Response to "Response to Reviewer TsWW"**
> >
> > Thank you for your response.
> >
> > I have noted the addition of the related work section and the additional experiment. Can you clarify, in response and in the paper, what the main difference is between the training strategy of Bansal et al. 2019, and the training strategy proposed herein? Both methods rely on estimation of collision loss and off-road loss, so the main difference really just seems to be whether the training is one-stage or two-stage. In the follow-up experiment of "ITRA (with infraction losses)", we get an improvement of performance w.r.t. "ITRA". But is there any explanation for the performance gain of TITRATED w.r.t. "ITRA (with infraction losses)"? What is the theoretical reason for why we should expect TITRATED to perform better? The way I understand it is that TITRATED is two-stage learning of p(S|A=true), and "ITRA (with infraction losses)" and Bansal et al. 2019 are both methods for one-stage learning of p(S|A=true) by removing density of p(S|A=false) through penalization during training.
> >
> > Also, unless I missed it, my third concern wasn't addressed. Please make clear how the data in Alg 1 is used in the training objective in Eq 10 through symbolic notation (my guess is that $\tilde D$ is supposed to be distributed by $p(s_{1:A}| A)$ and used only in the ${L}_{ELBO}$ term). This is critical to the clarity of the approach.

---

> > > ### Author Response · Authors · 2022-07-24
> > > **Re: Response to "Response to Reviewer TsWW"**
> > >
> > > We thank the reviewer for their follow-up questions. We updated the paper with Algorithm 2 which shows the final TITRATED training process. Please find below our response regarding the difference between TITRATED and Bansal et al. 2019.
> > >
> > > ITRA (Scibior et al. 2021), the foundation model used in this paper, is trained to imitate human driving behavior from human infraction-free demonstrations (INTERACTION dataset). The model yields an acceptable (but not zero) infraction rate when tested in the same locations from the training dataset. As shown in ChauffeurNet (Bansal et al. 2019), one can use various penalizing infraction losses to further improve the infraction rate performance while training the foundation model. These losses can be expressed as in Bansal et al. 2019 or the proposed infraction losses in our paper. Both infraction loss definitions aim to achieve the same goal but the gradient signal these functions propagate is different. Using these losses will improve the infraction rate performance further in the training set locations.
> > >
> > > In this paper, we are particularly interested in locations where human-generated realistic demonstrations are impossible to be obtained (i.e. in the CARLA simulator). All machine learning models, including ITRA and ChauffeurNet, suffer from degraded infraction rate performance when tested in out-of-domain (OOD) locations different from the ones in the training set. The goal of this paper is to obtain a model that produces fewer infractions while maintaining the human-likeliness that the foundation model learned by imitating human demonstrations. The main contribution of the TITRATED method is the amortization of the inference process that generates synthetic samples that satisfy certain conditions. In our case, these conditions are expressed as avoiding committing certain infractions. We generate these synthetic samples by rejection sampling from a given pre-trained probabilistic foundation model. Because these samples are generated by the human-like foundation model, we are able to obtain synthetic infraction-free demonstrations in these OOD locations that follow the approximated human behavior as learned by the foundation model. Finetuning the foundation model to yield a higher likelihood of these demonstrations (amortization) is the reason for the improved results in OOD deployment. We are not claiming that the model will necessarily generalize to other OOD locations, but rather provide a framework for improving the performance in the specified OOD locations used during the amortization process. Please refer to [this table](https://openreview.net/forum?id=M8D5iZsnrO&noteId=ozAwDEfgkI) to see the results of amortization.
> > >
> > > In addition to the amortization process, we use the proposed infraction losses to further reduce the infraction rates. The process of using such losses is shown both in ChauffeurNet (Bansal et al. 2019) and TrafficSim (Suo et al. 2021). Our proposed definition of these differentiable loss functions is different from both previous works and contrary to Bansal et al. 2019, we use the infraction losses to penalize randomly sampled trajectories generated by the amortized model under the prior. The TITRATED method is agnostic to the probabilistic foundation model used. Given a model that can generate trajectory samples of an (approximated) human-like behavior distribution, TITRATED can be used to improve its performance in the selected OOD locations. ChauffeurNet is a deterministic single-agent behavior model that will always produce the same single trajectory given the same initial conditions. This makes ChauffeurNet not suitable to be used as a probabilistic simulation model for the TITRATED method. On the other hand, models such as TrafficSim, MultiPath (Chai et al. 2020), and MFP (Tang et al. 2019) can benefit from the TITRATED method to further improve their infraction rate performance in OOD locations.

---

> > > > ### Comment · Reviewer_TsWW · 2022-07-29
> > > > **Re: Re: Response to "Response to Reviewer TsWW"**
> > > >
> > > > Thank you for your clarifications and additions. If I may summarize, the main point is that the method is for fine-tuning to situations without positive examples, but the ability to identify some negative examples. This separates it from prior methods that assume access to scenes with positive examples and a way to identify some negative examples, yet that have no way to incorporate a purely negative training signal (or at least, haven't tried) in some scenes. From this perspective, my concerns about experimental weaknesses are resolved. The evaluation shows improvement over using only data with positive and negative examples, but no fine-tuning to scenes without positive examples ("TITRATED" vs. "ITRA (With infraction losses)").

---

### Review · Reviewer_hW55 · 2022-07-04

**Summary Of Contributions:**

The authors propose a practical solution to the problem of transferring existing human behavior models to new environments without incurring unrealistic behaviors. The paper is motivated by the task of generating human-like behaviors in simulated agents for autonomous driving, specifically for CARLA. The authors remark that direct transfer of pre-trained models often results in unrealistic driving infractions, such as off-road driving and collisions. The proposed methodology, TITRATED, addresses this problem in two steps. First the data in the target environment is processed to remove any infractions predicted by a pre-trained behavior model using rejection sampling. Then, the pre-trained behavior model is fine-tuned on this filtered data with additional loss terms to discourage infractions. In this manner, the learned distribution over behaviors is pseudo-conditioned to not produce infractions in the new environment.

**Broader Impact Concerns:**

The authors provide a thoughtful consideration of the limitations of the TITRATED approach in the Discussion section, specifically, due to TITRATED inherently reducing or even eliminating infractions, dangerous situations may be missing from simulated scenarios. They also discuss the potential bias introduced by removing infractions from the data used for fine-tuning. In my view, this discussion is sufficient to address the intention of a separate Broader Impact statement, although it could be a nice addition if there is space in the paper.

**Requested Changes:**

My requested changes are based on the weaknesses described above. For more details, please see my response in the previous question.

1. Baseline against a version of ITRA fine-tuned on held out CARLA data without filtering out infractions to demonstrate the benefits of the proposed two-step TITRATED approach. (Critical)

2. Ablation studies: (a) only fine-tune with loss terms on CARLA data without filtering out infractions and (b) fine-tune on filtered data without the additional loss terms. (Would strengthen the paper)

3. Clarify the details and points of confusion listed in weaknesses above. (Highly encouraged)

4. Add a literature review section that clearly contextualizes the presented work in terms of existing literature in, for example, transfer learning and learning with constraints. Since the current manuscript is two pages under the page limit, there is sufficient space for this discussion. (Highly encouraged)

5. The references should be proofread (e.g., to ensure the year is not entered twice in a citation, conference proceedings are formatted consistently, the conference is listed instead of ArXiv whenever possible). (Nice to have)

**Strengths And Weaknesses:**

Strengths:
* Very well written paper. Easy to follow, concise, and polished.
* Well motivated by a practical issue in generating realistic simulated agents.
* The experiments were thoughtfully constructed and fairly extensive, supporting the claims made in the paper.
* The proposed solution is practical and intuitive.
* Fig. 2 is a highly effective illustration of the benefits of the proposed approach.
* The authors include a fair discussion of the limitations of the proposed TITRATED methodology.

Weaknesses:
* According to my understanding, the ITRA baseline was not fine-tuned on the CARLA data. If this understanding is correct, then it is unsurprising that the baseline does not do well and causes infractions upon transfer to a new setting, since as the authors mention, neural networks generally do not perform well on out-of-distribution data. To better showcase the benefits of the proposed methodology in producing fewer infractions in new settings, an ITRA baseline that is fine-tuned on held out CARLA data should be included in the comparison. If my understanding is not correct, and the ITRA baseline is fine-tuned on CARLA data, then this should be made more clear in the paper.
* There is no ablation study that illustrates the effect of the additional loss terms beyond simply fine-tuning on the filtered dataset. Similarly, there is no indication whether fine-tuning with the additional loss terms, without filtering the dataset, would also result in low infraction performance.
* Some design choices lack details or context:
    - On page 2, 'We generate infraction-free trajectories in target locations'. What about along the trajectory? According to my understanding from Eqs. (6) and (7), infractions are considered along the entire trajectory.
    -  On page 3, it was not entirely clear whether the road structure is included in the input to the network.
    - In the paragraph following Eq. (6), it is unclear what is the approximation made by Zhou et al. (2019) and consequently how the proposed use of this technique is novel in the behavior modeling context.
* Missing dedicated literature review. In particular, I would ideally like to see a discussion on transfer learning and learning with constraints.

---

> ### Author Response · Authors · 2022-07-18
> **Response to Reviewer hW55 (Part 1)**
>
> We would like to thank the reviewer for their feedback. We added a related work section in the revised manuscript as well as proofread all the references.
>
> Below are some clarifications to your questions.
>
> - **“On page 3, it was not entirely clear whether the road structure is included in the input to the network.”**
>
> The input to the network is an overhead birdview image ego-centered and ego-rotated as described in the paper [ITRA] of the foundation model (ITRA) used in this work. This birdview rasterizes the road structure and represents each agent as a rotated bounding box on top of the driving road surface. The ego agent has a limited field of view as described in ITRA.
>
>
> - **“In the paragraph following Eq. (6), it is unclear what is the approximation made by Zhou et al. (2019)”**
>
> Our collision loss is based on this open-source [implementation](https://github.com/lilanxiao/Rotated_IoU) of computing the differentiable IOU metric as described in Zhou et al. (2019). In general, computing an extremely accurate IOU value between two rotated bounding boxes is very difficult and laborious. This is because the intersection of two rotated rectangles is more complex to be determined. According to [1], It could be reduced to an empty set, to a point, to a rotated line segment, or to a rotated m-sided polygon, where m varies from 3 to 8. Instead, this implementation is using a brute force method based on [shoelace algorithm](https://en.wikipedia.org/wiki/Shoelace_formula) to approximately determine the overlapping area in a differentiable way.
>
> [1]: [Accurate IoU computation for rotated bounding boxes in R2 and R3, Abdelhamid Zaïdi, 2021](https://link.springer.com/content/pdf/10.1007/s00138-021-01238-x.pdf)
>
>
> - **“How the proposed use of this technique (differentiable IOU loss) is novel in the behavior modeling context”**
>
> To the best of our knowledge, we are the first to apply a differentiable IOU metric for rotated bounding boxes as a collision loss to directly minimize the overlapping area between two colliding agents. In contrast, the collision loss proposed by Suo et al., 2021 is distance-based and not area-based. The objective and interpretation of this loss function are different because the method is set to maximize the distance between two overlapping centroids. Each vehicle is represented by 5 centroids but in practice, this number depends directly on the vehicle size (i.e. a bus vs a small car). The default value might not be sufficient to ensure coverage over the whole vehicle. This complicates the usage of this loss function in practice whereas our proposed collision loss function does not suffer from this problem.

---

> > ### Comment · Reviewer_hW55 · 2022-08-01
> > **Response to Authors**
> >
> > Thank you for the clarifications, additional experiments, and including a new Related Works section. The experiment with ITRA fine tuned on CARLA and the discussion was particularly helpful, alongside the ablation studies. It is clear that 'fine tuning' in the regular sense is not possible in transfer from real behaviors to simulation. My concerns have been resolved!

---

> ### Author Response · Authors · 2022-07-18
> **Response to Reviewer hW55 (Part 2)**
>
> Below are the results of the requested experiments:
>
> - **“Baseline against a version of ITRA fine-tuned on held out CARLA data without filtering out infractions to demonstrate the benefits of the proposed two-step TITRATED approach.”**
>
> For this experiment, we fine-tuned ITRA on the infraction-free data produced by the CARLA autopilot including the infraction losses during training. Below are the results:
>
>
> |  | **Collision Rate** ${\times}10^{-4}$ | **Collision Rate**   ${\times}10^{-4}$ | **Off-road Rate**  ${\times}10^{-4}$ | **Off-road Rate**   ${\times}10^{-4}$ |
> |---|---|---|---|---|
> | **Scene name** | **TITRATED (All maps)** | **CARLA Autopilot Data** | **TITRATED (All maps)** | **CARLA Autopilot Data** |
> | Town01_Straight | 5.6 | 4.4 | 0 | 0 |
> | Town01_3way | 4.3 | 0.9 | 0 | 0.2 |
> | Town02_Straight | 3.4 | 2.1 | 4.9 | 1.3 |
> | Town02_3way | 3.1 | 3.1 | 0.5 | 1.5 |
> | Town03_Roundabout | 1.2 | 0.7 | 7.4 | 2.8 |
> | Town03_5way | 0.9 | 0.5 | 3.7 | 0.2 |
> | Town03_4way | 1.3 | 0.2 | 19.6 | 0.3 |
> | Town03_3way_Unprotected | 2.6 | 1.1 | 25.1 | 12.5 |
> | Town03_3way_Protected | 3.9 | 1.6 | 27.4 | 4.6 |
> | Town03_GasStation | 0.3 | 0.6 | 19.1 | 7.2 |
> | Town04_Merging | 0 | 0.6 | 0 | 3.4 |
> | Town04_3way_Large | 1.4 | 0 | 0 | 27.9 |
> | Town04_3way_Small | 3.4 | 0.2 | 0 | 0.2 |
> | Town04_4way_Stop | 2 | 0.4 | 0.09 | 4.5 |
> | Town04_Parking | 1.1 | 0.05 | 0.1 | 4.7 |
> | Town06_Merge_Single | 1.3 | 0.07 | 20 | 10.7 |
> | Town06_4way_large | 0 | 0 | 3.3 | 2.6 |
> | Town06_Merge_Double | 0 | 0 | 0 | 0 |
> | Town07_3way | 2 | 1.3 | 1 | 6.2 |
> | Town07_4way | 2.6 | 0.9 | 0.1 | 5.1 |
> | Town10HD_4way | 1 | 0.4 | 10.9 | 4.7 |
> | Town10HD_3way_Protected | 0.9 | 0.5 | 21 | 8.6 |
> | Town10HD_3way_Stop | 1.3 | 0.6 | 12.8 | 5.4 |
> | Average | 1.8 | 0.8 | 7.6 | 4.9 |
>
> Since we are able to train with all the provided initial cases due to the fact that we have infraction-free demonstrations provided by the autopilot, the infraction rates are further reduced. This is in agreement with our experiment in Section 3.1. The drawback of using autopilot data is that we now force our probabilistic model to learn from deterministically generated non-realistic demonstrations. This results in a degenerate probabilistic model that exhibits deterministic behavior. This is evidenced by the maximum final distance metric (MFD) [2] that measures the diversity in the predictions. The TITRATED model has an average score for all locations close to 3.4 whereas the model trained with autopilot data has an average score of 0.2.
>
> [2]: [Imagining The Road Ahead: Multi-Agent Trajectory Prediction via Differentiable Simulation](https://arxiv.org/abs/2104.11212)
>
> - **“Only fine-tune with loss terms on CARLA data without filtering out infractions”**
>
> For this experiment, we trained a TITRATED model with all the synthetic data by not applying inference (rejection sampling) to generate infraction-free trajectories.
>
>
> |  | **Collision Rate** ${\times}10^{-4}$ | **Collision Rate** ${\times}10^{-4}$ | **Off-road Rate** ${\times}10^{-4}$ | **Off-road Rate** ${\times}10^{-4}$ |
> |---|---|---|---|---|
> | **Scene name** | **TITRATED (No rejection sampling)** | **TITRATED** | **TITRATED (No rejection sampling)** | **TITRATED** |
> | Town01_Straight | 4.1 | 5.6 | 4.1 | 0 |
> | Town01_3way | 5.7 | 4.3 | 1.9 | 0 |
> | Town02_Straight | 3.4 | 3.4 | 0.4 | 4.9 |
> | Town02_3way | 4.3 | 3.1 | 1.5 | 0.5 |
> | Town03_Roundabout | 2.9 | 1.2 | 30 | 7.4 |
> | Town03_5way | 1.8 | 0.9 | 16.1 | 3.7 |
> | Town03_4way | 2.6 | 1.3 | 14.4 | 19.6 |
> | Town03_3way_Unprotected | 3.1 | 2.6 | 16.9 | 25.1 |
> | Town03_3way_Protected | 6.7 | 3.9 | 9.4 | 27.4 |
> | Town03_GasStation | 1.7 | 0.3 | 18.3 | 19.1 |
> | Town04_Merging | 0.1 | 0 | 3.8 | 0 |
> | Town04_3way_Large | 1.7 | 1.4 | 3.7 | 0 |
> | Town04_3way_Small | 3.1 | 3.4 | 1.5 | 0 |
> | Town04_4way_Stop | 2.3 | 2 | 0.7 | 0.09 |
> | Town04_Parking | 1.8 | 1.1 | 3.2 | 0.1 |
> | Town06_Merge_Single | 0.5 | 1.3 | 18.6 | 20 |
> | Town06_4way_large | 0.2 | 0 | 1.9 | 3.3 |
> | Town06_Merge_Double | 0 | 0 | 0 | 0 |
> | Town07_3way | 2.4 | 2 | 5.8 | 1 |
> | Town07_4way | 2 | 2.6 | 4.3 | 0.1 |
> | Town10HD_4way | 1.7 | 1 | 24.9 | 10.9 |
> | Town10HD_3way_Protected | 1.2 | 0.9 | 24.9 | 21 |
> | Town10HD_3way_Stop | 1.9 | 1.3 | 14.6 | 12.8 |
> | Average | 2.4 | 1.8 | 9.6 | 7.6 |
>
> This experiment highlights the importance of learning with synthetic data from the conditional target distribution (in this case infraction-free trajectories). The infraction losses used during training are helping the model to reduce infraction rates but training with infraction-free demonstrations will help reduce infractions further.

---

> ### Author Response · Authors · 2022-07-18
> **Response to Reviewer hW55 (Part 3)**
>
> - **“Fine-tune on filtered data without the additional loss terms”**
>
> For this experiment, we trained TITRATED without applying the additional infraction losses.
>
>
> |  | **Collision Rate** ${\times}10^{-4}$ | **Collision Rate** ${\times}10^{-4}$ | **Collision Rate** ${\times}10^{-4}$ | **Off-road Rate** ${\times}10^{-4}$ | **Off-road Rate** ${\times}10^{-4}$ | **Off-road Rate** ${\times}10^{-4}$ |
> |---|---|---|---|---|---|---|
> | **Scene name** | **ITRA** | **TITRATED (No infraction losses)** | **TITRATED** | **ITRA** | **TITRATED (No infraction losses)** | **TITRATED** |
> | Town01_Straight | 26 | 15.8 | 5.6 | 42.7 | 9.3 | 0 |
> | Town01_3way | 19.4 | 12 | 4.3 | 11.3 | 6.3 | 0 |
> | Town02_Straight | 11.2 | 9.4 | 3.4 | 51.5 | 16.5 | 4.9 |
> | Town02_3way | 10 | 9.9 | 3.1 | 33.9 | 4.2 | 0.5 |
> | Town03_Roundabout | 5 | 4.4 | 1.2 | 134 | 72.1 | 7.4 |
> | Town03_5way | 4.6 | 3.6 | 0.9 | 95.8 | 58.3 | 3.7 |
> | Town03_4way | 6 | 5.1 | 1.3 | 67.2 | 39.6 | 19.6 |
> | Town03_3way_Unprotected | 6.3 | 5.6 | 2.6 | 74.8 | 75.3 | 25.1 |
> | Town03_3way_Protected | 10.4 | 8.6 | 3.9 | 84.9 | 72.9 | 27.4 |
> | Town03_GasStation | 6.6 | 4 | 0.3 | 32.8 | 26.4 | 19.1 |
> | Town04_Merging | 0.2 | 0.1 | 0 | 47.8 | 7.2 | 0 |
> | Town04_3way_Large | 8 | 5.4 | 1.4 | 11.2 | 2.5 | 0 |
> | Town04_3way_Small | 16.6 | 13.5 | 3.4 | 29.6 | 14.9 | 0 |
> | Town04_4way_Stop | 10.9 | 8.1 | 2 | 32.4 | 21.3 | 0.09 |
> | Town04_Parking | 6.5 | 6.3 | 1.1 | 44.7 | 24.8 | 0.1 |
> | Town06_Merge_Single | 1.7 | 1 | 1.3 | 109.6 | 48.3 | 20 |
> | Town06_4way_large | 1.7 | 1.1 | 0 | 7.3 | 3.8 | 3.3 |
> | Town06_Merge_Double | 1.8 | 0.2 | 0 | 2.3 | 0.6 | 0 |
> | Town07_3way | 6.9 | 5.7 | 2 | 45.3 | 28 | 1 |
> | Town07_4way | 7.8 | 6.1 | 2.6 | 60.6 | 34.3 | 0.1 |
> | Town10HD_4way | 3.6 | 2.9 | 1 | 51.1 | 40 | 10.9 |
> | Town10HD_3way_Protected | 4.5 | 4.5 | 0.9 | 76.2 | 62.7 | 21 |
> | Town10HD_3way_Stop | 3.6 | 3.7 | 1.3 | 62.5 | 37.4 | 12.8 |
> | Average | 7.8 | 5.9 | 1.8 | 52.6 | 30.7 | 7.6 |
>
>
> This experiment shows that amortizing the inference process will produce models that are approximating the target conditional distribution. We additionally show in Section 3.1 that better inference will lead to lower infraction rates. Using the additional infraction losses helps to learn faster the infraction-free target distribution with less expensive inference algorithms such as rejection sampling with 10 trials.

---

### Review · Reviewer_34E5 · 2022-07-05

**Summary Of Contributions:**

In general, this paper aims at solving the problem of driving behavior modeling.
They present a new method that fine-tunes a pre-trained foundation model to novel road maps where human demonstrations are not available. Their target is to generate naturalistic driving behaviors without infractions such as collisions and off-road invasion.

Specifically, this paper has the following contributions:
* Propose a fine-tune pipeline to solve the OOD problem with little effort. Their method can quickly generate naturalistic driving behaviors in an unseen road map.
* Design two metrics to represent infractions of collision and off-road invasion. These two metrics are accurate and differentiable, which helps efficiently train their models.
* Conducted experiments to show that the finetuned method achieves less collision rate and off-road invasion rate. They also explore the influence of sampling methods on performance.


**Requested Changes:**

### Major concerns (critical):

* **Source of OOD.** I think the mismatch discussed in this paper is mainly caused by the map image. The bird-view image of CARLA is very different from INTERSECTION. However, this problem might be solved by training with broader datasets or using other representations than bird-view images. What's the opinion of the authors on this question?
* **Sampling is not efficient.** The reject sampling procedure seems trivial and too expensive as the authors said. Since this is the core part of the proposed algorithm, I doubt the proposed method is efficient. There could exist alternative methods to achieve this goal more practically and efficiently, for example, advanced sampling methods like MCMC and Multilevel splitting.
* **Unfair quality evaluation of behavior.** In section 3.2, the authors discussed how to evaluate the quality of human-like driving behavior. It is indeed difficult to evaluate whether a policy can generate naturalistic behavior. In the experiment part, the authors chose to use the distribution of acceleration as a surrogate metric, which I think may not be fair enough. The autopilot method used in CARLA is a very simple model, which usually takes a constant acceleration as input in most cases. Human drivers have very diverse behaviors, therefore have a wider range of acceleration. In addition, it is acceptable that ITRA has different distributions from human drivers due to the learning process, but the distribution of TITRATED is far away from the human driver and ITRA, indicating that TITRATED cannot generate naturalistic behavior (under the acceleration metric). I suggest the authors use a more complicated metric to evaluate the quality, for example, the likelihood of another generative model trained on very broad maps.
* **Results are not convincing.** In Table 2, the two metrics are represented with $\times 10^{-4}$, then the absolute difference seems very small. The ITRA method without finetuning seems already has a low collision rate and off-road rate.
* **Missing baselines.** The authors only compare ITRA and TITRATED in their experiments. Therefore, the conclusion heavily depends on the performance of the ITRA model. There are other methods that can generate the behavior of NPC in traffic scenarios, which should also be explored to investigate the importance of the foundation model. In addition, the authors may need to consider other baselines for dealing with the OOD mismatch, for example, using reinforcement learning methods to train agents on CARLA scenarios or using domain adaptation methods that are widely investigated in deep generative models.

### Clarifications and minor things:

* On page 2, “Alternatively, we can view this process as an instance of amortized inference, where the observations are known in advance, but the model is further conditioned, here on constraints.” Some explanations of amortized inference would help understand this sentence.
* On page 3, “… for each agent i and time t is obtained via differentiable rendering…” It is unclear why a differentiable renderer is necessary here. How does this renderer work? My understanding is that the bird-view image can backpropagate gradient only to x, y, and $\psi$.
* Provide more details about the ELBO?



**Strengths And Weaknesses:**

### Strengths:
* The paper is well-written and very easy to follow. Figure 1 helps understand the pipeline of the proposed method.
* The topic is interesting and the motivation is clear. Generating naturalistic behavior in simulations helps evaluate and train autonomous driving algorithms more efficiently.

### Weaknesses:
* The proposed sampling procedure is trivial and not efficient. More efforts need to be spent on the algorithm design to increase the contribution. Explained in the Requested Changes.
* Experiments are not through and some baselines are missing. Explained in the Requested Changes.

---

> ### Author Response · Authors · 2022-07-18
> **Response to Reviewer 34E5 (Part 1)**
>
> We would like to thank the reviewer for their feedback. Please find our response to each of your points below.
>
> - **Regarding the “Source of OOD”**
>
> We anticipate that a foundation model trained with more diverse locations and significantly more recorded (and varied) interactions between vehicles will reduce but not entirely eliminate the gap in performance between training and test locations. Even when a large dataset is available, it may not be practical to obtain data on maps similar to the test locations, for example when deploying in different countries or on unusual intersections. Obtaining these data can be prohibitively expensive and we believe that is beyond the scope of this paper.
>
> Instead, we argue that our proposed framework is generally applicable in the context of learning models that can efficiently generate conditional samples. Our method for generating synthetic conditional data from a foundation model and amortizing the generation process is useful for a number of tasks such as learning language models to generate sentences that contain no derogatory and offensive statements from a foundation general-purpose language model or generating molecules that have certain protein characteristics from a generative foundation molecular graph model. As long as a probabilistic generative model and a condition exist, TITRATED can be applied to efficiently learn to conditionally sample. For this paper, we picked the downstream task of conditioning human-like driving behavior models on avoiding yielding excessive infraction rates since we believe this to be a challenging, emerging, and under-explored problem in the literature.
>
> - **Regarding “Sampling is not efficient”**
>
> We agree that rejection sampling is often considered an inefficient inference algorithm and that more sophisticated inference procedures could be explored. MCMC methods and multilevel splitting would likely work better given a larger number of iterations, but since we only use 10 trials it’s unlikely that they can do better than rejection sampling for the same computational budget. We don’t use more trials to keep the computational costs under control, but we agree that this would be a good approach. In Figure 3 we show that better inference indeed can lead to better performance and in Section 4 we mention that finding better yet efficient inference algorithms is an open problem and part of future research.
>
> - **Regarding “Unfair quality evaluation of behavior”**
>
> Evaluating the realism of traffic simulations is mostly an open problem when human recordings are not present. To the best of our knowledge, the two established approaches [1] are user studies, where the participants are shown videos of real traffic replays and simulated traffic flow recordings and are asked to decide which one looks more realistic, or compare certain hand-picked traffic statistics. In our paper, we choose the latter, measuring realism by using the statistical characteristics of the acceleration response of all the behavior models used in the paper and the acceleration profile obtained from the INTERACTION dataset as a baseline against the non-human basic controller (autopilot) provided by the CARLA simulator. It is clear from Figure 4 in the paper that both the foundation model and the fine-tuned model yield a much smoother acceleration/deceleration response that is closer to human recordings than the autopilot, a deterministic rule-based model that always brakes aggressively right before committing an infraction.
>
> This experiment aims to (1) show that a TITRATED model remains close to the human-like acceleration response that both the foundation model and the human-recorded data yield, and (2) compare the TITRATED model to the standard deterministic rule-based autopilot model (autopilot) used to populate NPCs in the popular CARLA simulator. The idea of using the likelihood of a better-learned driving model to evaluate the human-likeness in CARLA is interesting and we will mention it as a future research avenue in the area of evaluating human-like models. Unfortunately, to the best of our knowledge, currently, we are not aware of a model that evidently produces realistic driving behaviors in CARLA that's publicly available and could be used for this purpose.
>
> Furthermore, we have multiple recordings of various simulated traffic flows that further demonstrate the ability of our models to avoid infractions intelligently and not pick trivial solutions (i.e., stopping or driving backward to avoid infractions). These video recordings will be available after the double-blind period to avoid the potential unmasking of the authors.
>
> [1]: [Dictionary-based Fidelity Measure for Virtual Traffic](https://ieeexplore.ieee.org/document/8481568)

---

> ### Author Response · Authors · 2022-07-18
> **Response to Reviewer 34E5 (Part 2)**
>
> - **Regarding “Results are not convincing”**
>
> The foundation model is already a very good driving behavior. We believe though that the infraction rates it yields in OOD locations are not acceptable for practical deployment given the critical nature of the collision and off-road constraints. The challenge we are facing is the ability to reduce these rates to an acceptable level without sacrificing the human-likeliness of the foundation model while applying it in domains where it is impossible to obtain human-recorded data (i.e. CARLA simulator). According to [1], the human collision rate is approximately 1 collision every 100,000 miles. The table below shows the collision rate per 100,000 miles for both the foundation (ITRA) and our proposed model (TITRATED). Although TITRATED significantly reduces the number of collisions compared to ITRA, it is still not better than human drivers highlighting the importance of this research.
>
> | **Scene name** | **ITRA** | **TITRATED** | **TITRATED (All maps)** |
> |---|---|---|---|
> | Town01_Straight | 14200 | 1093 | 3059 |
> | Town01_3way | 7885 | 1951 | 1748 |
> | Town02_Straight | 2995 | 749 | 909 |
> | Town02_3way | 2910 | 961 | 903 |
> | Town03_Roundabout | 1260 | 429 | 303 |
> | Town03_5way | 1334 | 464 | 261 |
> | Town03_4way | 2827 | 1178 | 613 |
> | Town03_3way_Unprotected | 3007 | 1289 | 1241 |
> | Town03_3way_Protected | 5384 | 2537 | 2019 |
> | Town03_GasStation | 2481 | 226 | 113 |
> | Town04_Merging | 95 | 48 | 0 |
> | Town04_3way_Large | 13494 | 3880 | 2362 |
> | Town04_3way_Small | 16600 | 6400 | 3400 |
> | Town04_4way_Stop | 8838 | 3811 | 1622 |
> | Town04_Parking | 4715 | 3410 | 798 |
> | Town06_Merge_Single | 368 | 238 | 282 |
> | Town06_4way_large | 1638 | 289 | 0 |
> | Town06_Merge_Double | 1585 | 441 | 0 |
> | Town07_3way | 2035 | 649 | 590 |
> | Town07_4way | 2379 | 702 | 793 |
> | Town10HD_4way | 531 | 251 | 148 |
> | Town10HD_3way_Protected | 904 | 322 | 181 |
> | Town10HD_3way_Stop | 529 | 279 | 191 |
> | Average | 4261 | 1374 | 937 |
>
> [1]: [Rates of Motor Vehicle Crashes, Injuries and Deaths in Relation to Driver Age, United States, 2014-2015](https://aaafoundation.org/rates-motor-vehicle-crashes-injuries-deaths-relation-driver-age-united-states-2014-2015/)
>
>
> - **Regarding “Missing baselines”**
>
> Most of the established domain adaptation methods are not applicable in this setting, since we don’t have human trajectories available in target locations. The main obstacle to using reinforcement learning is that we don’t know how to define a reward component encouraging human-like behavior, especially since in ITRA (and most of the other deep generative models for predicting human trajectories) likelihood is not tractable. Additionally, RL is likely to require significantly longer training than the imitation learning we use, which is already pushing the limits of what’s practical. As for using different foundation models, they are all qualitatively similar to ITRA and known to suffer from excessive infraction rates [2]. Including such additional models would take a lot of work and we don’t see any reason why they would produce qualitatively different results, acknowledging that the magnitudes of infraction rates might vary between models. Once again, we emphasize that the effort (both computational and human) required to incorporate such additional baselines is substantial and we do not see proportional benefits from including them.
>
> [2]: [TrafficSim: Learning to Simulate Realistic Multi-Agent Behaviors](https://arxiv.org/abs/2101.06557)

---

### Decision · Action_Editors · 2022-08-09

**Recommendation:** Accept as is

**Comment:**

The paper presents a method that fine tunes and adapts a foundational model to the situations where there is missing data. While done in the autonomous driving setting, the paper makes an important contribution to using the available and incomplete data in the space of the safety critical applications.

The authors have made significant improvements to the original manuscript, addressing all the reviewers comments. The ablation studies and positioning working within the related works are especially helpful. As a result, the reviewers agree and I concur that this work would be welcome contribution to the research community and might inspire future work in this important problem space.